# How Do Gendered Labour Market Trends and the Pay Gap Translate into the Projected Gender Pension Gap? A Comparative Analysis of Five Countries with Low, Middle and High GPGs

**Gijs Dekkers** [1,2,*], **Karel Van den Bosch** [1,3], **Mikkel Barslund** [4], **Tanja Kirn** [5], **Nicolas Baumann** [5], **Nataša Kump** [6], **Philippe Liégeois** [7], **Amílcar Moreira** [8] and **Nada Stropnik** [6]

[1] Belgian Federal Planning Bureau, 1040 Brussels, Belgium; kvdb@plan.be
[2] Centre for Sociological Research (CeSO)—KU Leuven, 3000 Leuven, Belgium
[3] Herman Deleeck Centre for Social Policy, University of Antwerp, 2000 Antwerp, Belgium
[4] HIVA Research Institute for Work and Society—KU Leuven, 3000 Leuven, Belgium; mikkel.barslund@kuleuven.be
[5] Center of Economics, University of Liechtenstein, 9490 Vaduz, Liechtenstein; tanja.kirn@uni.li (T.K.); nicolas.baumann@uni.li (N.B.)
[6] Institute for Economic Research, 1000 Ljubljana, Slovenia; natasa.kump@ier.si (N.K.); stropnikn@ier.si (N.S.)
[7] Luxembourg Institute of Socio-Economique Research (LISER), 4366 Esch-sur-Alzette, Luxembourg; philippe.liegeois@liser.lu
[8] Institute of Social Science, University of Lisbon, 1600-189 Lisbon, Portugal; amoreira@iseg.ulisboa.pt
[*] Correspondence: gd@plan.be; Tel.: +32-(0)25077413

**Abstract:** This article explores how the Gender Pension Gap (GPG)—the relative difference in average pension received by men and women—might evolve in the future in various European countries, given past, current, and projected future labour market behaviour and earnings of women and men, and current pension regulations. The GPG reflects career inequalities between women and men, though these are partly mitigated by the redistributive impact of the public retirement pensions. They are further mitigated by survivor benefits. This study aims to document both mechanisms in the projections of the GPG. As the GPG varies widely across European countries, we analyse countries with a high (Luxembourg), high and low middle (Belgium and Switzerland Portugal), and low (Slovenia) GPG. We find that the GPG will fall significantly in all five countries over the coming decades. The fundamental drivers behind this development are discussed. In addition to the base scenario, we simulate two variants to show the impact of the Gender Pension Coverage Gap and of survivor pensions. Additionally, we project the GPG if current labour market gender gaps were to remain at their present level, and, conversely, if these were to disappear overnight. These alternative scenarios, one of which also serves as a robustness test, suggest that the future decline of the GPG is largely the result of labour market developments that have already happened during the past decades.

**Keywords:** Gender Pension Gap; dynamic microsimulation

## 1. Introduction

In the EU, men on average receive a pension that is 30% higher than the pension received by women. In some member states, this Gender Pension Gap (GPG) is even substantially higher.[1] The GPG is receiving increasing interest from policy makers and other stakeholders. Understanding how the GPG will develop in the coming decades and its main drivers is therefore of policy interest.

This paper discusses the expected development of the GPG in several EU countries and Switzerland and explores how gendered labour market trends and developments

translate into these projected developments of the GPG. As the GPG varies widely across Europe, ranging from 44% in Luxembourg to less than 1% in Estonia, with an average of 29.5% (European Commission and Social Protection Committee 2021b, p. 104), we study the evolution of the GPG in European countries with a low, middle, and high GPG: Luxembourg, which has comparatively high GPG (44%), Belgium (31.9), Switzerland (33.1), and Portugal (28.2), which are countries with an average GPG, and Slovenia (16.4), a country with a comparatively low GPG (Figure 1).

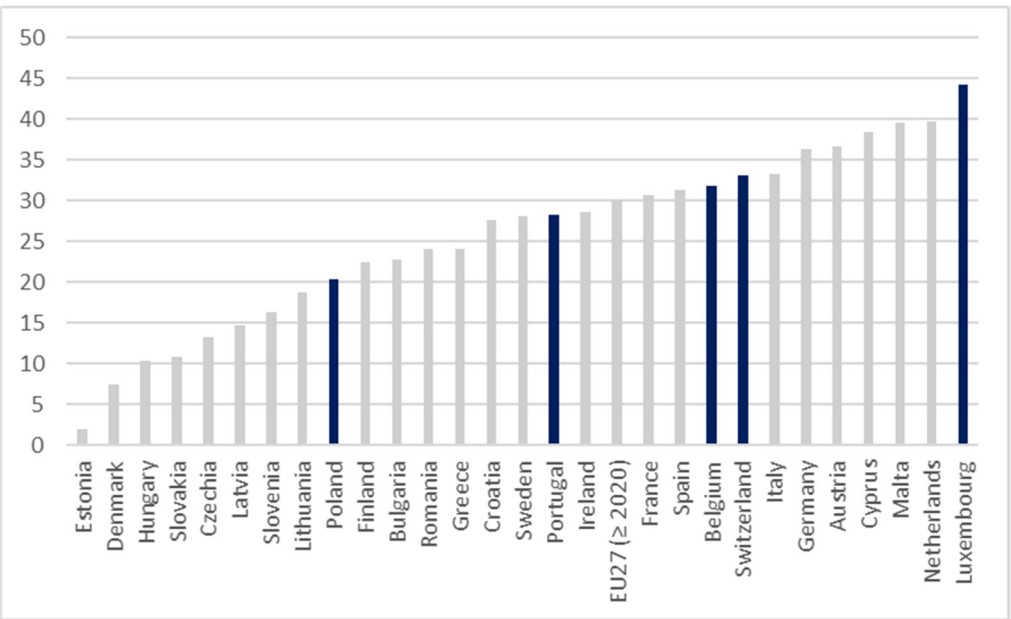

**Figure 1.** GPG in European Union member states and Switzerland (data for 2019). Source: European Commission (2022). All aged 65 and more. The dark bars describe countries for which simulation results are presented in this paper.

In the standard measure of the GPG, people with zero pensions (overwhelmingly women), as well as everyone below age 65 are excluded from the calculation. The GPG is usually higher than the gender pay gap. This is mainly due to the fact that the GPG reflects the much larger gender labour market inequalities in past decades. As pensions are in most systems a direct or indirect function of previous earnings and periods worked, between-gender inequalities in work interruptions, career lengths at retirement, and part-time work, exacerbated by the resulting wage-scarring, accrue and are reinforced over a person's lifetime (Veremchuck 2020; Jolly 2014, p. 50; Bettio et al. 2013; Lequien 2012; Staff and Mortimer 2012; Thévenon and Solaz 2013; Möhring 2018). Another element that makes the relation between earnings and the later pension benefit complex is the redistributive character of first-pillar pension systems. As a result, the relation between on the one hand the earnings gap and differences in labour market participation, and on the other hand the GPG is not linear.

We use dynamic microsimulation to project the GPG as a function of (1) the gender differentials in past labour market behaviour of currently active people, (2) the gender differentials in prospective labour market behaviour of currently active people, as well as of future entrants into the labour market, and (3) the pension systems as they were in 2019. The microsimulation models are based on large datasets from administrative sources for Belgium, Luxembourg, and Slovenia, on EU-SILC 2013 for Portugal, and on EU-SILC 2018 for Switzerland. Through alignment techniques we make sure that the simulation of future labour market behaviour and future gender gaps are consistent with demographic and macro-economic projections. For Belgium, Luxembourg, Portugal, and Slovenia, the projections by the European Union's Ageing Working Group[2] (AWG) are used. These projections are published in the Ageing Report (European Commission 2020, 2021b), which

assesses the long-term sustainability of public finances and the economic consequences of ageing populations of the EU Member States. For Switzerland, the simulations are aligned with projections by the Swiss Federal Statistical Office.

Alignment to these projections enhances the realism and credibility of our results, and also helps to make them comparable across countries. An important limitation is that the simulation models can only handle first-pillar pensions in the four countries of Belgium, Luxembourg, Portugal, and Slovenia; second-pillar occupational pensions are only covered in the model for Switzerland.

We find that the GPG will fall significantly in all five countries over the next two decades. In Slovenia and Portugal, the gap will be close to five percent already in 2030 and will have essentially disappeared in 2040. In Belgium and Luxembourg, the gap is reduced to seven and five percent in 2050, respectively, which is less than one-third of their 2020 levels. In Switzerland, for which results will be presented separately below, the decrease of the GPG is smaller and it would end up at 13 percent.

In addition to a base scenario with the standard GPG definition, we project two variants of the latter and present results for two alternative labour market scenarios. In the first variant, we include older people with zero pensions, which implies that the resulting GPG in fact incorporates the Gender Pension Coverage Gap, which measures the extent to which older women have less access to the pension system than older men (European Commission and Social Protection Committee 2018, p. 76). In the second variant of the GPG, survivor pensions are excluded, which illustrates the important impact of this pension component on the GPG. In the first alternative labour market scenario, gender gaps in employment and other labour market characteristics are kept at their current level from 2020 on. It turns out that the GPG would decline almost as much as in the base scenario. Conversely, when simulating that gender gaps in the labour market (including the wage gap) would disappear overnight, the GPG would decrease not much more than in the base scenario, and only in the longer term.[3] Finally, this study simulates the combined impact of equalising both the employment rates and average earnings between men and women from 2021 on. Besides serving as a robustness test, this scenario confirms the expectation that the eradication of the main sources of gender-inequality would over time reduce the GPG to zero in those countries where this does not happen in the base scenario. All scenarios furthermore confirm that these developments are very gradual in their impact on the GPG.

There are few existing studies that project Gender Pension Gaps. Halvorsen and Pedersen (2019) also use a microsimulation model to study the distributional effects of the reformed Norwegian pension system and to project the GPG for one cohort (individuals born in 1963). We project the GPG for all cohorts and several countries. Bonnet et al. (2006) use a dynamic micro-simulation model to study the effects of France's 1993 and 2003 pension reforms and show that these tend to slow down the narrowing of the GPG. Their analysis is limited to private-sector pensions. Additionally, like Halvorsen and West-Pedersen, they limit their analysis to some cohorts; in this case those born between 1965 and 1974. Finally, Chłoń-Domińczak (2017) develops a "Forward-Looking Gender Pension Gap Index". This is a multidimensional measure that is essentially a weighted sum of changes of three employment-gap indicators (employment gap, pay gap, and work intensity gap) and four qualitative indicators of "pension system compensation" (career break compensation; pension redistribution; pension indexation, and retirement). The weights used to combine these indicators are chosen by the author "to reflect expert assessment of the risk posed by selected indicators on the future gender pension gap" (Chłoń-Domińczak 2017, p. 9). The advantage of this approach is that no complex models are required, but the disadvantage is that the results ultimately are a function of the subjective choices of indicators and weights.

The remainder of the article is organised as follows. In Section 2, we give an overview of the socio-economic context in the countries under scrutiny. Section 3 then presents the relevant pension regulations of each country. Section 4 details the methodological approach of dynamic microsimulation and the assumptions made therein, while Section 5 is devoted to the presentation of results. Section 6 concludes.

## 2. Socio-Economic Context

### 2.1. Gender Gaps in the Labour Market

An individual's pension outcome in many countries is a complex function of the labour market career, the earnings trajectory, and pension accumulation during possible periods of unemployment or absence from the labour market. Hence, differences in pension outcomes among specific individuals, say a man and a woman, can be due to several factors that are often difficult to disentangle. However, generally speaking, when assessing gender differences in pension outcomes at the macro level there are three main drivers of pension outcomes: gender differences in employment histories and in earnings, and the redistributive elements in the pension systems.

### 2.2. Gender Difference in Employment Rates

The traditional male breadwinner family model has been on the decline in Western Europe over the past 50 years (Bonnet et al. 2012). In Belgium, Luxembourg, and Portugal, women at prime working age are in 2018 substantially more likely to work and thereby to accumulate their own pension entitlements than around 1980. Figure 2 below shows the observed employment rates for men and women aged 15–64 between 1983 and 2018, and the projected values produced for the Ageing Working Group sustainability projections (European Commission 2021b) for the EU countries in this study, and from the Swiss Federal Statistical Office for Switzerland (FSO 2020, 2021). Belgium, Luxembourg, and Portugal all had gender employment gaps of more than 30 percent in the mid-1980s. In 2018, this gap has narrowed to between six and eight percentage points. For Slovenia, comparable data are only available since 1995, but these suggest that the employment gap was smaller in the past and has not declined much. Switzerland has the second highest employment rate for women aged 15 to 64 (76.3%) compared to the EU27 countries; the employment rate for women was already rather high in the early 1990s, and it has converged to that of men in the last three decades.

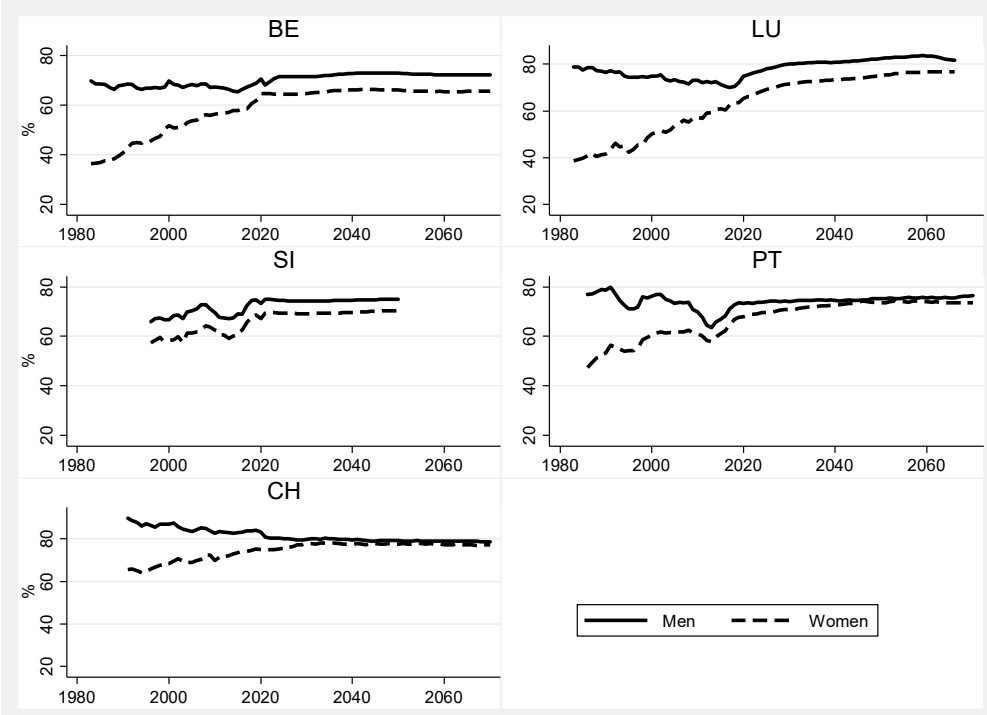

**Figure 2.** Employment rates for men and women (15–64, 1983–2070). Sources: EU-LFS; Swiss Federal Statistical Office, AWG projections; national reports. "BE" stands for Belgium, LU for Luxembourg, SI for Slovenia, PT for Portugal, and CH for Switzerland.

In all five countries, the employment rates of men and women have converged over the past decades. The Ageing Working Group is, however, projecting a slowdown in convergence, except for Portugal where employment differences will be negligible by 2050 and in Switzerland where complete convergence will be reached by 2030. Nevertheless, past convergence will continue to influence the future Gender Pension Gap.

### 2.3. Gender Difference in Wages

The second main driver of pension income is earnings during the work life. As is the case with employment rates, earnings differences between men and women tend to become smaller in many countries (Blau and Kahn 2017; Goldin 2014), although this development is sometimes slow (Fransen et al. 2012). Figure 3 shows the development of the gender pay gap in the five countries as well as the EU27 as a whole.[4] In the EU27, the pay gap stands at 14.1% in 2019 and has decreased only marginally from its peak value of 16.4 in 2012. The pay gap shows a consistent decrease in Luxembourg and Belgium, while it was increasing in Slovenia between 2009 and 2018, and in Portugal until 2015, and is virtually stable at a high level in Switzerland.

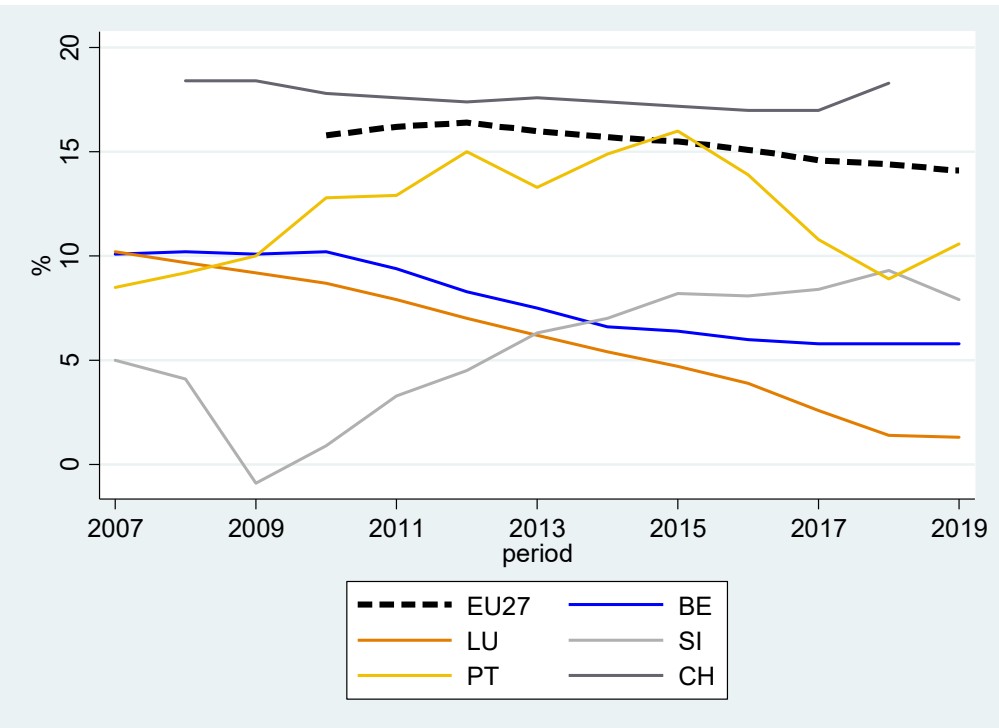

**Figure 3.** Gender pay gap (hourly wages) in four countries, 2006–2019. Source: Eurostat Gender pay gap in unadjusted form [SDG_05_20] 24 February 2021; retrieved 3 December 2021.

Harmonised data on gender differences in wages are not available for longer time periods, but Figure 4 illustrates the development in Belgium since 1972. This indicator is better suited to show the impact of earnings differences on pensions, because it shows the difference in gross earnings per month (i.e., not correcting for part-time work) in manufacturing. It is considerably higher, but, more importantly, it shows that the decline in the Gender Pay Gap so defined over the long-term is substantial.

The AWG does not make projections of the gender wage gap. There is reason to believe that the gender pay gap will be reduced further as (1) women's educational attainment levels are rapidly catching up to and even surpassing that of men (van Hek et al. 2016), and (2) the gap is lower for younger age groups (European Commission 2021a; OECD 2012). However, it remains unclear whether it is set to disappear given the fact that women still see a substantial decline in wages relative to men throughout their career,[5] and assuming

that gender-specific industry patterns and part-time employment rates do not change dramatically. In our base-scenario, we therefore have chosen a conservative approach in assuming that current gender inequalities in pay remain unchanged in the projection period. Past gender wage differences and changes herein across cohorts are still represented in the model by differences in the stock of accrued pension rights, in the same way that past differences in employment rates are also embodied in the projections.

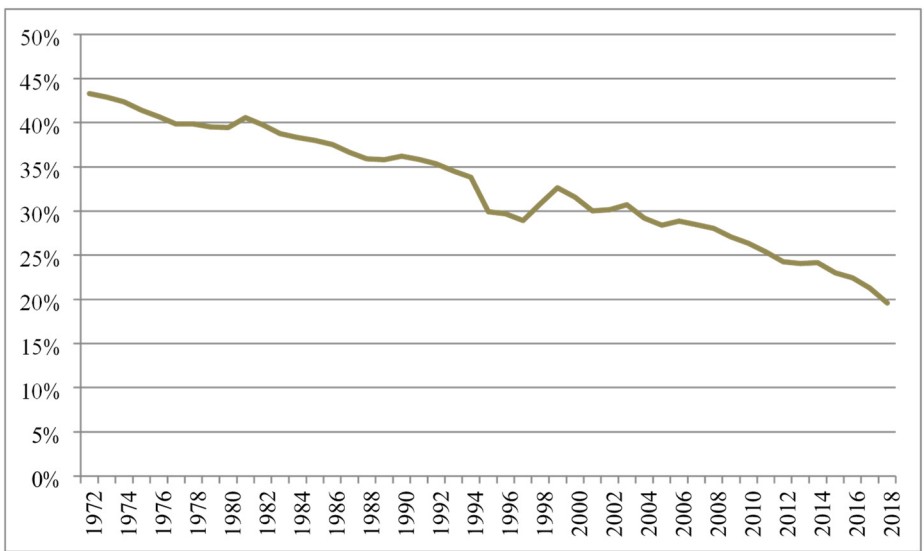

**Figure 4.** Gender pay gap (monthly wages) in Belgium, 1972–2018. Source: Institute for the Equality of Women and Men (2021), De loonkloof tussen vrouwen en mannen in België. Rapport 2021., Figure 11, p. 21. Note: Gender pay gap in terms of gross earnings per month in manufacturing, so not correcting for part-time work.

*2.4. Recent Evolution of the Gender Pension Gap*

Eurostat measures the Gender Pension Gap on the basis of EU-SILC since the early 2000s. Figure 5 shows a modest decrease of the GPG for the EU27 as a whole for the 65+ from 35% in 2010 to 30% in 2018. Among the five countries in this study, the GPG ranged from 44% in Luxembourg to 16% in Slovenia, with Switzerland (38%), Belgium (32%), and Portugal (28%) lying in between. Also in a wider European perspective, the GPG is very high in Luxembourg, low in Slovenia, and close to the EU average in Belgium, Portugal, and Switzerland.[6] The GPG has been nearly halved in Slovenia and seems to also have declined in Portugal. In the other countries there is no clear trend in any direction.

The standard reported GPG ignores those older persons that do not have a pension at all. One should evaluate the level and evolution of the GPG in connection with the Gender Pension Coverage Gap, which measures the extent to which women, compared to men, have their own independent access to pension system benefits, i.e., the difference between the percentages of women and men aged 65+ receiving any pension. The Gender Pension Coverage Gap is highest, but decreasing rapidly, in Belgium (Figure 6). In Slovenia, the official Gender Coverage Pension Gap figures suggest that women are more likely than men to have any pension benefit. This is mainly because disability pensions in Slovenia are not converted into the old-age pensions at the SRA. If disability pensions were included, then the gap would amount to 0.8% in 2018 (Kump and Stropnik 2021, p. 12). In Luxembourg and Portugal, the Gender Pension Coverage Gaps are small and do not show a clear trend, and in Switzerland it is close to zero due to the universal coverage of first-pillar pension. One reason for the high Gender Pension Coverage Gap in Belgium is the system of the household rate: married people (mostly women) can forego their low pension, so that their partner (mostly their husband) receives the more generous household rate, if that results in a higher pension for the couple. Note, finally, that this system removes many low pensions for women, which would otherwise increase the Gender Pension Gap.

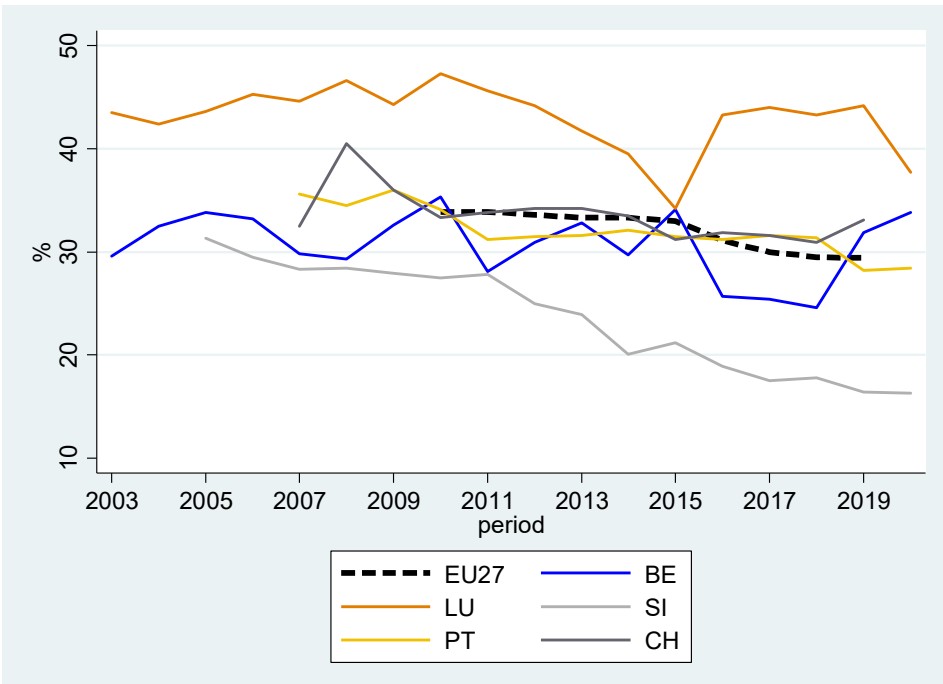

**Figure 5.** Recent evolution of the Gender Pension Gap (%), 2003–2020. Source: Eurostat, EU-SILC, table reference ilc_pnp13, last update: 19 November 2021, extracted on 2 December 2021. Note: Though EUROSTAT does not indicate a break in series, the series for Belgium may be broken between 2018 and 2019 due to a change in data collection.

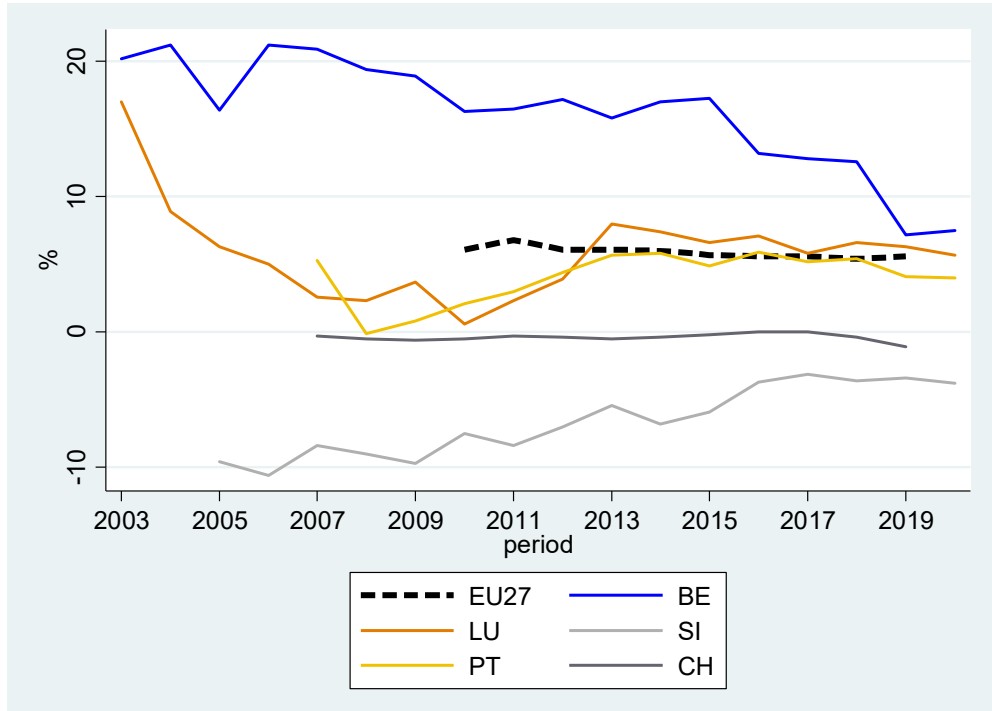

**Figure 6.** Gender Pension Coverage Gap (%). Source: Eurostat, EU-SILC, table reference ilc_pnp14. (65+) last update: 2 December 2021, extracted on 3 December 2021. Note: Though EUROSTAT does not indicate a break in series, the series for Belgium may be broken between 2018 and 2019 due to a change in data collection.

### 3. Pension Regulations

The selected countries not only differ with respect to the GPG, but also allow a comparison between countries with Bismarckian-type pension systems (Belgium, Luxembourg, Portugal, Slovenia) and Beveridge-type systems (Switzerland). The two systems can be roughly distinguished on the basis of the insurance obligation, the benefit target, and the financing. Bismarck-type pension systems cover persons in gainful employment and aim to replace income and maintain the status acquired during working life. Beveridge-type pension systems cover the entire population, but provide often only a basic pension, which is financed out of taxes or tax-like contributions, often supplemented with well-developed occupational pensions and individual (pension) savings (Ebbinghaus 2021). Since second-pillar occupational pension incomes are a large component of the Swiss system, and data are available, these are included in the results presented below. Unfortunately, this was not possible for the other countries, partly because they are less centrally regulated (and therefore harder to simulate), partly because of lack of data.[7] The right panel of the following Table 1 shows the coverage rate by gender, by pension system (Public, Occupational, Personal), persons aged 65+; the left panel shows the same for the active population (aged 16–64), based on the SHARE survey.

**Table 1.** Coverage rates of pension pillars.

| Country | Active Persons | | Older Persons (Men/Women) | | |
|---|---|---|---|---|---|
| | Occup | Personal | Public | Occup | Personal |
| BE | 59.6 | 38 | 98.8/83.1 | 2.4/1.5 | 3.3/1.5 |
| LU | 5.1 | .. | 96.4/92.0 | 9.5/1.6 | 4.2/1.7 |
| SI | 36.5 | 1.4 | 90.2/87.6 | 1.1/0.7 | 1.8/2.1 |
| PT | 3.7 | 4.5 | 87.2/78.8 | ../.. | 0.8/1.0 |
| CH | 56.8/43.2 | .. | 97.6/98.7 | 82.9/69.5 | 45.1/34.9 |

Source: Right panel; European Commission, 2018 PAR, Table 5, p. 72 (various years between 2006–2015). Data for Switzerland: FSO Switzerland[8] Left Panel: idem, Table 6, p. 79 (2016). Data for Switzerland: FSO Switzerland.[9] The rate of coverage of the first pillar for the active population is 100% in all countries. .. — not available.

This table shows that the coverage of the occupational second pillar varies between nearly 4% in Portugal and almost 60% in Belgium. For the 65+, the coverage is considerably lower, which shows that occupational pensions are a growing phenomenon.

In both Bismarckian- and Beveridge-type systems, the years of contributions as well as the contributions paid are crucial for determining pension entitlements (Leitner 2001). However, interruptions or reductions in employment affect pension entitlements differently (Arza 2007; Arcanjo 2019). Since in Bismarckian-type systems the obligation to contribute is linked to employment (or to employment-related states such as unemployment), interruptions in employment lead to missing years of contributions, which in turn may lead to lower pensions. In Beveridge schemes (in their ideal type), on the other hand, there is a general obligation to contribute, so that periods without employment lead to lower contributions (if, for example, only a minimum contribution is paid), but no contribution years are lost. However, since Beveridge-type systems often aim only at providing subsistence and thus first-pillar pension income often consists of flat-rate benefits, which are highly redistributive (Kolmar 2007; Cremer and Pestieau 2003), the pension losses associated with a reduction in employment are comparatively small. In contrast, in Bismarckian-type systems, which aim to replace earned income in retirement, lower contributions are associated with stronger pension reductions, and longer career and higher earnings will generally result in a higher pension. However, this relationship is mediated by several distributive elements in the pension systems, such as minima, ceilings, and imputed contributions or earnings.

The remainder of this section gives a brief and partial overview of pension regulations in each of the five countries covered.[10] The focus is on the elements in the pension system,

which are relevant for understanding the simulation results. Save for some minor rules involving the retirement age and survivors' benefit in Switzerland, the pension systems of all countries in this study are gender-neutral. Nevertheless, the description of the various systems is relevant because they determine how the gender differentials in pay and career length filter through to pensions. Finally, for a better understanding of the context of the study, Table A1 in the Appendix A to this paper presents the average replacement rates as well as legal retirement ages, today as well as in the future, following current legislation. The statutory retirement age for someone entering the pension system is currently 65 years in Slovenia and Luxembourg; although currently 65, it will be 67 years in Belgium from 2030 on and expected to be 68 years in Portugal, where the statutory retirement age is linked to the development in life expectancy. In Switzerland, the retirement age is 65 for men and 64 for women.

In some countries there are different first-pillar pension schemes for different sectors. In Portugal, Slovenia, Luxembourg, and Switzerland, the general pension scheme covers both employees and the self-employed. In Belgium and Luxembourg, civil servants have their own scheme, while in Belgium there is also a separate system for the self-employed. The projections for Belgium cover all these schemes. The pension system for new civil servants in Luxembourg is very similar to the general system for employees. Therefore, results for Luxembourg also apply to civil servants. Finally, in Portugal, there is a sub-system for civil servants enrolled up to 2005. In addition to these contribution-based schemes, there are also non-contributory means-tested social assistance schemes, which serve as a social safety net. In Belgium, there is a special such scheme for people older than the statutory retirement age, which is included in the total pension in our projections. In Switzerland, recipients of a pension can claim a supplementary benefit, which is included in the first-pillar pension in the projections. In the other countries, older people with insufficient income can apply to the general social assistance scheme, but this assistance is not treated as part of the pension. Finally, disability pensions in Slovenia continue to be paid after the SRA. This is why, in contrast to the other countries, the simulations for Slovenia include the disability pension benefits for older persons as well.

In the Belgian employees' scheme, the normal accrual rate is 1.33% applied to wages earned during the career and adjusted only to current prices, which is equivalent to a 60% replacement ratio after a full career of 45 years. If a pensioner is the head of household with a dependent spouse, the accrual rate is increased to 1.67%. The accrual rate is higher for low wages due to the existence of minimum pensions (applied only in case of a career of at least 30 years) and a minimum claim per working year. Conversely, the accrual rate is lower for high wages, as these are subject to a ceiling for the pension calculation. The self-employed scheme is similar, except for a much lower replacement rate. Civil servants enjoy higher pensions, because these are based on the wages during the last 10 years of work. In all schemes, there is no qualifying period; pension entitlements are built up from the first day of work.

Pensions in Luxembourg's system are a sum of four components: a flat rate component, corresponding to a percentage of the social minimum income (24.4% in 2020); a pro-rata enhancement, i.e., a percentage of the total contributory income (1.8% in 2020); an incremental enhancement that depends on the sum of the individual's age plus the total of contributory years; and an end-of-year allowance bonus. For the initial calculation of the pension, wages are revaluated with respect to prices and the real wage evolution. To be eligible for an old age pension, a person must have accumulated a total of at least 10 years of contributory periods.

In Portugal, the pension is calculated on the average monthly salary of the 40 years with the highest earnings (adjusted by the Consumer Price Index). There is a qualifying period of at least 15 years. The annual accrual rate varies by earnings bracket between 2% for the highest earnings and 2.3% for the lowest earnings. There is a minimum pension, which increases with the length of the career, though less than proportionally. There is also a social old-age pension granted to people who do not meet the career length condition. A

social supplement is granted to bring the pension up to the guaranteed minimum amount, but (until recently) access to it was subject to very restrictive conditions.

In Slovenia, there is a minimum contributory period of 15 years. The annual accrual rate is set to 1.36% for each year after 15 years (for the first 15 years, the total accrual rate is 29.5%) and for 40 contributory years reaches 63.5% of the pension base. The latest changes in legislation, which introduced gender-neutral pension rules, were implemented in 2020 with a transitional period until 2025, during which the accrual rates remain higher for women than men. The pension base comprises wages during the 24 most favourable consecutive years of insurance and is subject to a minimum and a maximum. Additionally, a minimum pension is granted to those with full careers.

The first-pillar pension in Switzerland is a pay-as-you-go system, in which a full pension is received after 44 years of contributions (though contribution gaps can be paid retroactively for up to 5 years). The level of the pension depends on the average (updated) earned income during the career, but with a minimum and a maximum (the latter is twice the minimum, but still below the at-risk-of-poverty threshold). For married couples, earnings are added up and split across the spouses. The joint pension of married couples cannot exceed 150% of the individual maximum pension. However, as long as only one partner in a couple is retired, he or she gets the full individual amount. The second pillar of the Swiss pension system is a funded system and targets, in combination with the first pillar, a pension income of about 60 percent of the average lifetime labour income. It consists of a mandatory and a voluntary part. The mandatory part applies to all employees of 17 and older to the part of their annual salary that is between 21,510 CHF and 86,040 CHF.[11] There are no official statistics on average contribution rates, therefore only the mandatory part of the Swiss second-pillar pension system is simulated.

All five countries have some compensating mechanism such that pension rights are (at least partly) accumulated during periods of unemployment and disability, but these vary in generosity. In Luxembourg, Switzerland, and Slovenia, pension rights accrual is compensated only for a limited period and at the level of the unemployment benefit, while in Belgium the pension accrual during unemployment is based on past earnings and not limited in time. The basis for calculating pension rights during unemployment in Portugal is the level of unemployment benefit. In Switzerland, when one becomes unemployed or disabled, the accrual of old-age pension continues as before in the first pillar, but then based on a lower income (the unemployment or disability benefit itself). In case of unemployment, the second pillar no longer covers old age but only death (i.e., survivors' benefits) and disability. As a result, periods of unemployment and disability result in gaps in retirement assets through the second pillar. The unemployment benefit is limited in time; if someone no longer receives unemployment compensation, the contributions to the first pillar become due as a non-employed person, while contributions to the second pillar can be made voluntarily and missing contributions can be paid at a later date.[12]

The first-pillar pension systems in the countries in this study are redistributive in the sense that pension relative to previous earnings tends to be lower when the latter are higher. These redistributions are affected through floors and minimum pensions which drive up the lowest pensions, and ceilings in earnings that push down the highest pensions. An indicator of the degree of redistribution so achieved is the difference in the (gross) theoretical replacement rates (TRR) between men with a low-earning career (earning 2/3 of the average wage) and men with a high-earning career (starting off at average wage and progressing linearly and ending at double the average wage). This amounts to 27, 29, 17 and 7 percentage points in Belgium, Luxembourg, Portugal and Slovenia, respectively (figures for projected gross pensions in 2056).[13] It is clear that these redistributive elements reduce the GPG, as the group of older persons with lower pensions is dominated by women (Herd 2009).

Besides old-age pensions, our analysis includes survivors' pensions. These have a strong gender-dimension and a non-trivial impact on the GPG. Because women live longer than men and because many women tend to have lower old-age pensions than men,

women are overrepresented among the recipients of survivor pensions. The generousness of survivor pensions varies across countries. Here, we limit the descriptions to the most common situation where both the deceased and the surviving partner are already retired. In Belgium, the survivor pension in the employees and self-employed schemes is equal to 80% of the deceased person's old-age pension if the latter had a dependent spouse (i.e., a spouse with no income), and 100% otherwise. Anti-cumulation rules put a ceiling on the combination of a survivor pension with an old-age pension. In Luxembourg, the survivor pension amounts to 100% of the flat-rate elements of the deceased spouse's pension, and 75% of the proportional elements. The survivor pension that is in excess of a threshold can be reduced by 30% if it is combined with other incomes. Portuguese widows and widowers inherit 60% of the pension of the deceased spouse (unless the latter had more than one spouse over the course of their life). In Slovenia, the surviving partner has a choice between receiving 70% of the pension of the deceased partner, or his or her own retirement pension, topped-up by a survivors' supplement of 15% of the full survivor pension. In Switzerland, the eligibility requirements for widows differ from those for widowers. Older women are entitled to a survivors' pension when they have children or when they were married for at least five years. Men are entitled only if at the time of widowhood, they had children below 18 years. The first-pillar widow's and widower's pension are equivalent to a maximum of 80% of the retirement pension. If someone can claim a first-pillar pension at the same time as the widow's or widower's pension, only the higher pension is paid. The pension of the second pillar for surviving dependents amounts to 60 percent of the retirement pension to which the insured person would have been entitled.

Finally, because of different life expectancies between men and women, the indexation regime that applies to the pensions of those in retirement can affect the GPG. If pensions are not or incompletely indexed, the pensions of the oldest pensioners will tend to fall behind those of younger cohorts. As women live longer than men, this can increase the Gender Pension Gap. In Belgium, Luxembourg and (in part) in Portugal, pensions are indexed by reference to consumer prices, but there are different rules regarding real increases on top of this. In Belgium, the old-age pensions of civil servants are automatically adjusted to an increase in the real wage of working civil servants. Real increases in the employees and self-employed schemes are the result of a political choices within a specified budget; for the projection we assume that minimum pensions are increased by 1% per year and other pensions by 0.5% per year. In Luxembourg, as long as revenue from contributions exceeds the system's expenditure, pensions in payment are fully readjusted to the real wage evolution. When this condition is not satisfied, the readjustment mechanism is to be reduced by at least 50% or even cancelled. In Portugal, indexation in real terms of pensions depends on GDP growth and is more generous for lower pensions than for higher ones. In Slovenia, pensions are indexed to 60% of the increase in the average gross salary and to 40% of the average increase in the cost of living. Switzerland uprates first-pillar pensions at least every two years by the average of the increases of prices and wages. The second pillar pension is a fixed amount throughout the remaining life years.

There are several specificities in the pension systems which can have implications for our model results. In Slovenia, only the best 24 years of earnings (uprated annually by the average nominal wage growth) serve as the basis for the final pension assessment, and in Portugal the best 40 years are taken into account. Second, in some pension systems, the accrual rate per year is increased if people continue to work after a certain age and/or after they have exceeded a given number of years of contribution. In Slovenia, a very generous accrual is available after 40 years of contribution: 1.5% per six months for a maximum of three years. The pension system in Luxembourg also provides some bonus accrual to encourage extending the working life. Women generally have more unstable work histories than men, and therefore are less often able to access such bonus accruals, which tends to increase the GPG (Frericks et al. 2009).

## 4. Methodology and Pension Models

Eurostat defines the Gender Pension Gap (GPG) as the difference between the average pension received by men and women in percent of the average pension received by men. The average is defined for the population who are above the age of 65 and are receiving a pension. The GPG is based on gross pensions, including first-pillar pensions, second-pillar pensions and pensions from private pension plans. Because of limitations in the starting data or in the microsimulation model, it is not possible to project second- and third-pillar pensions. As indicated above, therefore, our simulations are limited to first-pillar pensions, except for Switzerland.

### 4.1. Variant GPG Definitions and Alternative Scenarios

While our focus in this article will be on the GPG following the Eurostat definition, this is not the only interesting way to measure the difference in pension received between men and women. For this reason, we present projections for two variant GPG definitions, whose function is primarily to shed more light on the GPG in the reference scenario. First, the GPG based on average pensions of the 65+ by itself paints only half the picture. For example, the gender gap in pensions excludes those that do not receive a pension benefit at all. The proportion of 65+ not having a pension is considerably higher for women than for men (Burkevica et al. 2015, Figure 4; also Figure 6 above). A straightforward way to take account of the gender old-age coverage gap in the GPG is to simply include those people without a pension as receiving zero pension, and then recalculate the GPG for the whole 65+ population. As the sum of pensions by gender would not change, and more older women than men have zero pensions, including these in the calculation will increase the GPG. Using SILC data, we found that the impact of this alternative definition is especially important for Belgium. This study will discuss how the projections of the GPG change if men and women with zero pensions are included.

Second, pensions include both old-age (or retirement) pensions and survivor pensions. Survivor benefits are especially important for women. As we will show, a variant GPG based on old-age pensions only is much higher than the standard GPG, indicating that survivor pensions are highly redistributive between genders, and dampen much of the differences in average pensions between men and women.

In addition to the base labour market scenario projected by the AWG, we simulate two alternative scenarios. In the first one, gender gaps in employment and other labour market characteristics are kept at their current level from 2020 on. This shows how much of the future decline in the GPG is due to labour market developments that have already happened. In the second alternative scenario, we simulate the development of the GPG assuming that gender gaps in the labour market would disappear overnight.

### 4.2. Microsimulation Modelling Framework

A projection of future pensions outcomes requires knowledge of the accrual of pension rights in the current and future population, which depend on past employment and earnings up until today, and on assumptions about future employment rates and wages. Microsimulation models that incorporate a detailed specification of the pension system can combine these elements to project future pensions of men and women and to quantify the contribution of these elements to reducing or widening the existing and future GPGs. We briefly discuss the microsimulation framework and data sources below.

A dynamic microsimulation model simulates the behaviour of micro-units over time, which means that it allows to show the impact of labour market decisions and pension system characteristics on the individual level. Starting from a dataset of individuals grouped in households, these models simulate the necessary demographic and labour market characteristics that are relevant for accrual of pension rights. Thus, individuals find and lose jobs, earn wages or receive a benefit, and build up a pension; households are formed and dissolved. Finally, men and women retire and receive a pension, and they die, after which their surviving spouse may be eligible to a survivors' pension. In that,



the models construct longitudinal projections of the individuals in such a way that their pension outcomes reflect the characteristics and provisions of the pension systems in force in the various participating countries.

Thus, microsimulation models are designed for the simulation of the impact of exogenous societal economic or policy changes on the income distribution, including poverty risks and inequality. They can easily handle complex and non-linear schemes, such as pension systems and progressive tax systems. They also allow to simulate the impact of labour market trends among various groups on pension outcomes later in life. This is why dynamic microsimulation models are used regularly to simulate pensions and pension reform. Finally, since the models simulate the entire distribution of pension outcomes, it is possible to look beyond the standard GPG and calculate several GPG variants. Appendix B to this paper presents a more formal framework for dynamic microsimulation.

A key element in the microsimulation models used in this paper is the degree to which they rely on auxiliary information, in the form of projections from other models, and policy hypotheses. For example, the AWG-projections include the employment rate by age and gender in any projection year. The goal of the microsimulation exercise is to create the longitudinal profiles of individuals and their households in such a way that these employment rates are reproduced by the model. To make the (micro) outcomes consistent with external macro projections, such as those produced by the Ageing Working Group, the micro simulation models make use of alignment-techniques (Dekkers et al. 2010). This technique involves using micro-level behavioural risk profiles (usually in the form of survivor, logit, or probit models) in combination with auxiliary transition probabilities. This ensures consistency between micro and macro data and has been used for projections of the future at-risk-of-poverty rate for pensioners for the Pension Adequacy Report (European Commission and Social Protection Committee 2018).[14] The micro simulation models in this study follow this approach by simulating the various projected Gender Pension Gaps while aligning to the 2021 AWG projections and hypothesis (for Belgium and Slovenia) or the 2018 projections (for Portugal and Luxembourg). This includes population projections, labour force projections (participation rates, employment, unemployment), wage growth rates, and hypotheses for the uprating of pensions and social security benefits. For the Belgian model, the semi-aggregate model MALTESE uses the 2021 AWG projections to provide more specific projections to MIDAS, including employment in the private and public sector, civil servants, but also various specific social security categories. In using this auxiliary data, our work fits into the earlier work by Dekkers et al. (2015a, 2015b, 2018), who project various indicators of pension adequacy using AWG projections and hypotheses.

A key element in any microsimulation model is the dataset of individuals and households on which it is based. This is typically a sample of individuals and their households in some starting year, which also includes retrospective information on these individuals insofar as it is relevant for the simulation of the pension benefit. The model then takes this data forward, and, at retirement, the pension benefit of those that were halfway through their career in the starting dataset is simulated on a mixture of observed and simulated careers and earnings. The Belgian model MIDAS, the Luxembourg model MIDAS_LU and the Slovenian model DYPENSI are all based on large samples of administrative data. In Belgium, it is a sample from the resident population in 2011, linked to administrative labour market, social security, and fiscal data and data from an administrative Census. The sample is roughly 550 thousand individuals, which is about 5% of the population in 2011. The Luxembourg model runs on a dataset of roughly 530 thousand individuals, which is 100% of the population of Luxembourg in 2016. DYPENSI runs on an administrative sample of nearly 113 thousand individuals, which is about 5% of the population of Slovenia in 2007. The starting dataset for the Portuguese model, DynaPor, is the 2013 wave of EU-SILC. This survey dataset comprises about 14 thousand individuals, which is roughly 0.13% of the population. As EU-SILC does not provide information on contributory careers, administrative data were used to impute for each individual aged 15 to 75 the average contributory career and average reference remuneration of individuals (adjusted for inflation)

with the same age, gender, work status, and earning decile. Finally, input for MIDAS_CH, the Swiss model, is the EU-SILC data for 2018, which has a sample size of approximately 15 thousand persons, or about 0.18% of the population. This dataset also lacks information on the past career. Hence, contrary to the other country models, this implies that complete careers have to be simulated before pensions can be calculated. As a result, projections for Switzerland could only be made for the year 2070, when the cohort for whom complete careers could be simulated had reached retirement. For further details on the datasets and pension definitions, we refer to the national reports.

Dynamic microsimulation also has its downsides. First of all, the observed simulation results are a function of many different and interacting behavioural equations on the level of the individual as well as the household, of which the cumulative effects in the long-term are difficult to assess, even though they are strongly and deliberately constrained by the alignments. Secondly, as is the case for most models of this kind, the models used in this paper do not allow for within-period feedback effects or general equilibrium modelling.

## 5. Results

### 5.1. Base-Scenario Results

The developments in the Gender Pension Gap until 2070 for five countries in the base scenario are shown in Figure 7. All five countries will see a strong decline in the GPG over the next 50 years. The decrease is particularly steep in Slovenia and Portugal where the gap will be close to 5 percentage points already in 2030. In Belgium and Luxembourg, the reduction is more moderate, but the GPG is still more than halved in the period until 2050. In Switzerland, the GPG will be reduced to 13% in 2070, which is about a third of the 2020 levels.

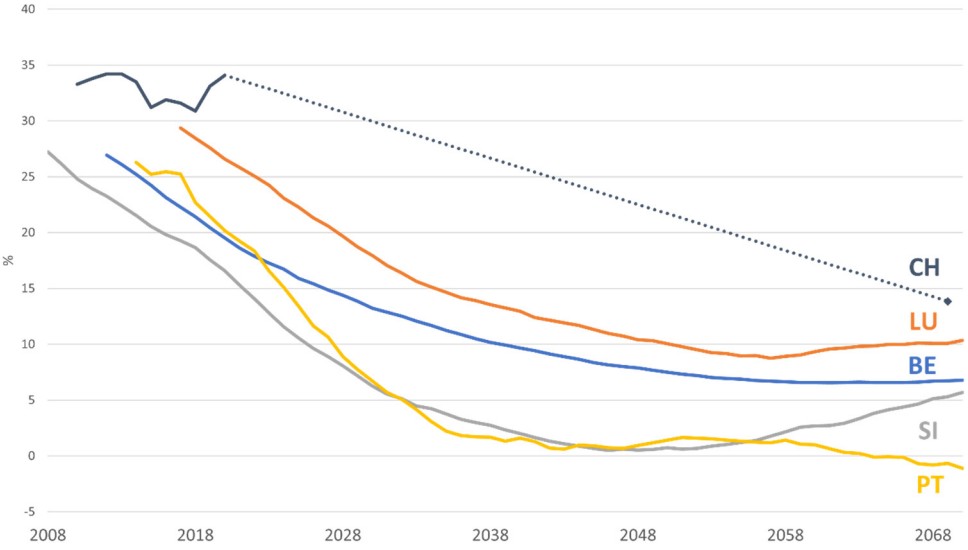

**Figure 7.** Gender Pension Gap under Ageing Working Group assumptions and projections (2008–2070). Source: Projections by MIGAPE country teams, Eurostat. Note: Since the starting data refer to different years, the starting years of the projections vary by country. Belgium: 2012; Luxembourg: 2017, Slovenia: 2008; Portugal: 2015. Data for Switzerland 2010–2020 are not simulated but show the EU-SILC observations for those years. For Switzerland, due to reasons explained in the main text, no intermediate results between 2020 and 2070 are available.

Except for Slovenia, the projected declines seem much steeper than those observed in EU-SILC, as shown above in Figure 5. One reason is that by their very nature, dynamic microsimulation models project long-term developments and are not subject to sample fluctuations. More importantly, as indicated above, these projections are based on statutory pensions only, and do not include complementary (occupational and/or private) pensions. For Belgium at least, there is evidence that the importance of those complementary pensions

has increased, and also that the Gender Gap in those incomes is greater than in the first-pillar pensions (Conseil Supérieur des Finances 2021).

While many country-specific factors, including the pension systems, play a role in these evolutions, some patterns can be recognised. The most important factors driving the decline in the GPG are the sharply decreasing employment gaps in Belgium, Luxembourg, and Portugal in recent decades (see Figure 2 above). Narrowing gender pay gaps are also likely to play an important role. Also in Slovenia, the decreasing and low GPG is driven by high women's activity rates during the latest decades (Figure 2) and especially the increasing activity rates of women at higher ages (55 and over); higher educational attainment and consequently higher salaries of women; and a lower number of women receiving survivors' pensions. In Switzerland, the decline is primarily driven by a decline in the difference in the number of hours worked between men and women.

Furthermore, the AWG projections assume that part-time work rates remain constant. An important reason that the GPG stays at a higher level in Belgium and Luxembourg (and partly, Switzerland) than in Portugal and Slovenia is that in the former countries there is a large gender difference in the part-time working rate: 27 percentage-points in Belgium and 25 percentage-points in Luxembourg, while part-time working rates are low for both genders in Portugal and Slovenia.[15] In the projections for Switzerland, it is assumed that women will increase their work percentage.

Note that the GPG in Slovenia is projected to increase again near the simulation horizon. Birth cohorts of women currently aged 45 and over do not show a gender gap in employment rates, which means that they will retire with completed pension contribution periods very similar to those of men from the same birth cohort. However, in younger age groups, women currently do have lower employment rates than men. In simulation, even if their employment rates will catch up later in life, this means that these younger cohorts of women will enter into retirement (from 2050 on) with comparatively shorter careers, and hence be confronted with a higher GPG. In Luxembourg, finally, the GPG increases again also near the simulation horizon, albeit only a little. This is due not to old-age benefits, but rather to survivors' benefits, which in projection, will be less focused on women than today, partly because life expectancies of men and women will have converged. Furthermore, the convergence of earnings between men and women will reduce the proportion of women receiving a survivors' pension towards the level of men. Hence, the downward impact of the survivors' pensions on the GPG will become smaller (but not disappear) over time. As retirement pensions of women will even at the simulation horizon remain lower than those of men, this reduced impact of the survivors' pensions will drive the GPG back up again.

### 5.2. Variants and Alternative Scenarios

This section will expand the discussion of the base-scenario results by presenting additional results for two variant definitions of the GPG: a GPG that incorporates the Coverage Gap, and a GPG without survivor pensions that shows the impact of the latter. Additionally, two alternative labour market scenarios will reveal the contribution of the AWG-projected employment rates and relative wages to the projected GPG. In the first one, current labour market outcomes are kept constant at their 2020 level, while in the second one gender labour market gaps are assumed to be closed from 2021 onwards.

#### 5.2.1. GPG and the Coverage Gap

The standard GPG does not include zero pensions. Therefore, the European Commission complements this with the "gender gap in pension coverage", which measures the extent to which women have less access to the pension system than men (European Commission and Social Protection Committee 2018, p. 71). As a sensitivity test, we have calculated a variant where the standard GPG and the gender gap in pension coverage are combined into a single indicator, simply by basing the GPG on the average pension benefit including zero values. Since the coverage gap of the 1st pillar is almost zero in Switzerland,

the GPG including zero pensions is not calculated for Switzerland as it would be similar to the standard GPG.

Around 2020, the variant GPG with zero values is considerably higher than the standard GPG in Luxembourg (14.5 pp) and Belgium (about 8.3 pp) (Figure 8). In Slovenia, the difference is only 3.3 pp. This is due to the fact that the coverage gap is higher in Belgium and Luxembourg than in Slovenia (see Figure 6). Over time, both in Belgium and Slovenia, as the proportion of older people without a pension decreases, the GPG including zero pensions decreases faster than the standard GPG, and at the end of the projection period, convergence is nearly complete. The results in Luxembourg show a convergence that is comparatively small, as the coverage gap, which is already fairly stable in the period 2005–2020 (see Figure 6), does not fall as much as in other countries. The reason for this is the minimum contributory period for an old-age pension of 10 years in Luxembourg. Even though this minimum can be achieved with part-time work with few hours,[16] it does affect women more than men and therefore prevents the coverage gap from falling to zero. In Portugal, where the coverage gap is already low, including the zero pensions has virtually no impact on the projected GPG.

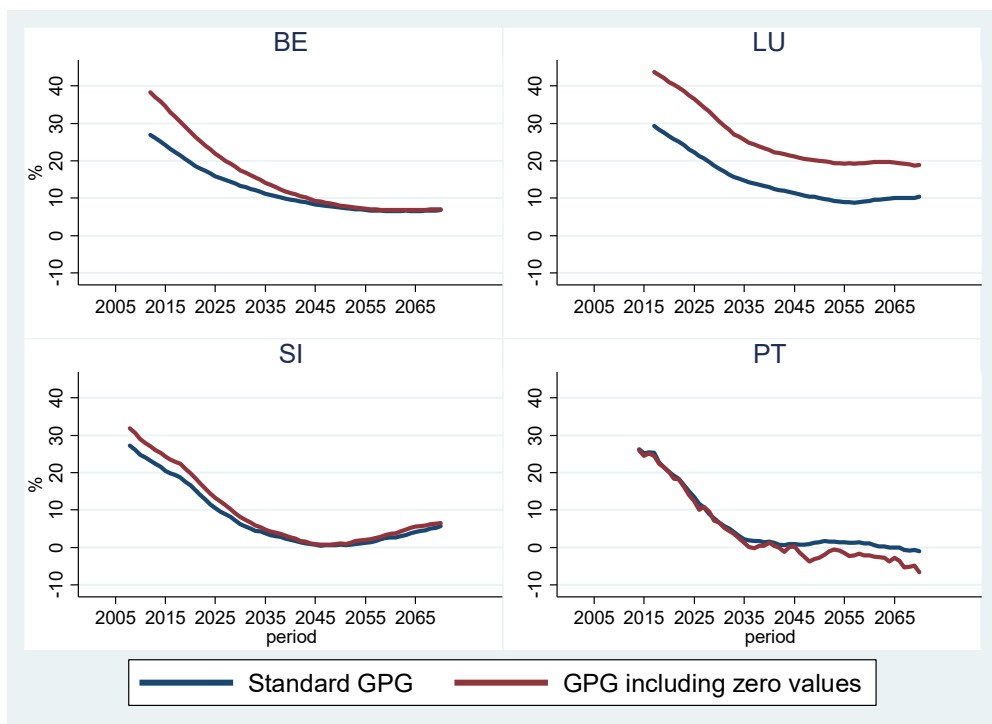

**Figure 8.** Impact of zero values on the Gender Pension Gap (2008–2070). Source: Projections by MIGAPE country teams; results for CH are not available. Note: see Figure 7.

### 5.2.2. Impact of Survivors' Pension

The importance of the survivor pensions on the GPG outweighs its importance in terms of expenditures.[17] In all four countries, individuals whose partner (or ex-partner) is deceased are eligible to a survivors' pension benefit (see Section 3 above). Eligibility conditions differ between the countries.[18] However, because of common patterns of longer life expectancy for women, the fact that women tend to be younger than their husbands, and finally the cumulation rules with old-age pensions, most beneficiaries of survivors' benefits are women. Moreover, since women who are or were married have on average shorter labour market careers compared to men and to women who never married, the survivors' benefit is an element in the pension system that is decreasing the GPG (Bonnet et al. 2006). This is confirmed by Figure 9 where the standard GPG is shown together with a projection of the alternative GPG without survivors' benefit.

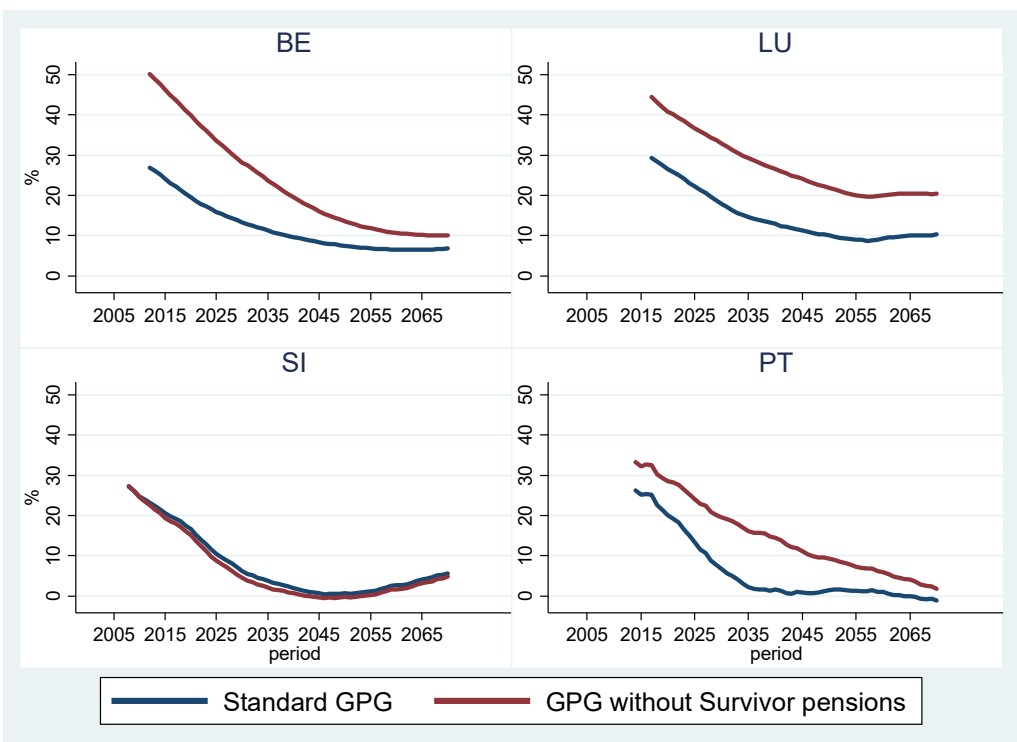

**Figure 9.** Impact of survivors' pension on the Gender Pension Gap (2008–2070). Source: Projections by MIGAPE country teams; results for CH are not available. Note: see Figure 7.

In Belgium and Luxembourg, the impact of the survivors' benefit is comparatively large today, reducing the GPG by around 20 and 15 percentage points, respectively. This is also because the system is more generous in these countries than in Portugal and Slovenia, in that expenditures per beneficiary on survivor benefits are proportionally closer to the expenditures per beneficiary of old-age pensioners in Belgium and Luxembourg than in Portugal and Slovenia (European Commission and Social Protection Committee 2021b, Figure 10 (upper panel), p. 36). In Belgium, the impact declines over time, as new cohorts of widows more often than before have an old-age pension of their own, and the ceiling on the sum of the old-age and survivor pension implies that they get no or only a small survivor pension as a result. In the other countries, the correction of the survivors' benefit in case of an old-age benefit is less severe. In Luxembourg, survivor pensions can be reduced by only 30% (and there is a generous threshold) when combined with an old-age pension. For Portugal, the reduction is smaller at somewhat less than 10 percentage points, but their role will increase in the future, and without survivor pensions the GPG would not be eliminated in 2040. The impact of survivor pensions in Slovenia, though small, is the opposite of that in the other countries: excluding these pensions leads to a smaller, instead of a larger GPG. The main reason is the higher employment rate among women in Slovenia in the past, which implies that they have often a substantial old-age pension of their own. This works in combination with the comparatively stringent anti-cumulation rules: the full means-test of lower retirement benefits in the survivors' benefit implies that many widows receive a relatively low survivors' supplement to their own pension (see Section 3 above), while many other women with a full survivor pension drop out of the calculation of the alternative GPG, as they have no other pension.

### 5.2.3. GPG When Assuming That Current Labour Market Outcomes Are Kept Constant

The gender pension gap reflects past differences between men and women in the prevalence of part-time work, unemployment, withdrawal from the labour market, and the pay gap, which accrue over a person's lifetime. In that sense, the GPG is a backward-looking indicator of past inequalities, and the simulated GPG in the reference scenario

is therefore the result of developments that (i) took place before the starting year of the simulation (in the "past"), and (ii) expected developments that are projected by the AWG and are included through the alignments. The question is which of these developments is the most important in their impact on the projected GPG. One way to answer this question is to look at what will happen with the GPG if we assume that current labour market outcomes—employment rates and relative wages—are held constant at their current level until the simulation horizon.[19] We refer to this as the *Constant* scenario. Figure 10 shows the AWG projection together with a Constant scenario project until 2070. The base scenario following the AWG projections produces a smaller GPG, but there is only a small difference between the Base and Constant scenarios. The difference is mainly due to the assumption in the AWG projection of convergence in employment rates for men and women in the age group 55 to 64 years old (European Commission 2020, p. 48), and for Switzerland, that the gender gap in part-time employment will decrease. The long-term Constant scenario illustrates that gender differences in labour market outcomes today are consistent with a GPG of between one and eight percentage points, and that the projected decline of the GPG is mainly the result of labour market developments that have already taken place. In other words, the finding that the difference between the results in the constant scenario and the reference scenario are small suggests that the decline in the GPG according to the reference scenario is mainly the result of labour market trends that took place before the starting years of the various models.

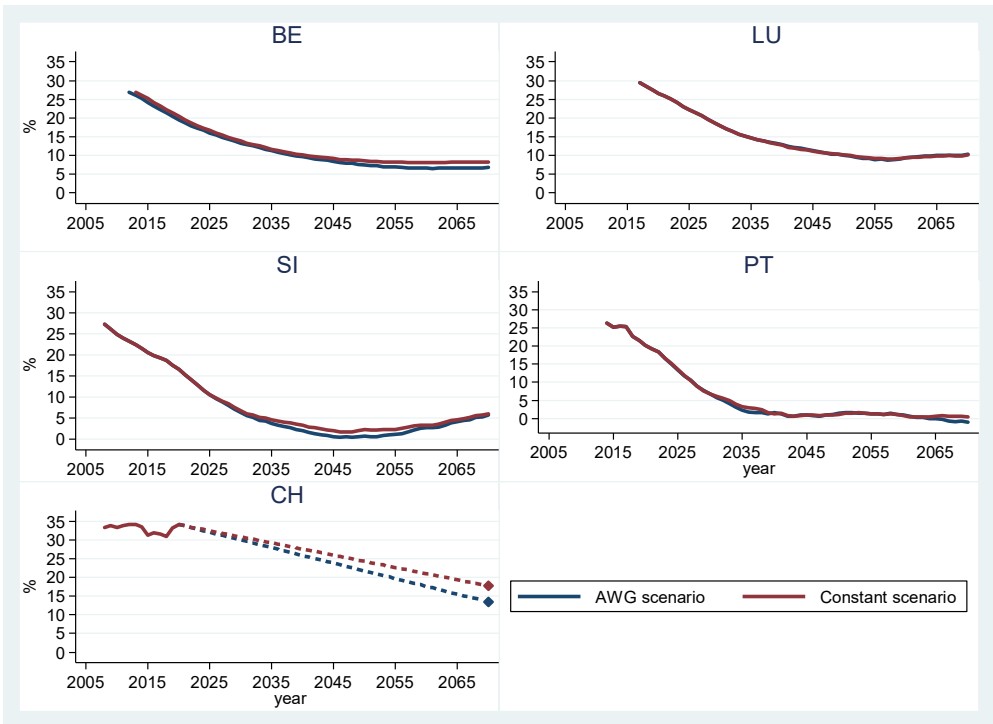

**Figure 10.** Gender Pension Gap (standard definition) under the constant scenario (2008–2070). Source: Projections by MIGAPE country teams, Eurostat. Note: see Figure 7.

In conclusion, according to our simulations and taking into account the AWG-projections, the GPG would decrease significantly over the next two decades, even if current labour market outcomes were to remain constant in the future. In the next section, we consider the impacts of overnight equalisations of labour market outcomes between men and women on the future path of the GPG.

5.2.4. Gender Pension Gap Projections When Equalising Labour Market Status and Pay

In this section, we analyse how the projected GPG would change if gender differences in labour market status, e.g., employment, unemployment, inactivity, and in the gender pay gap would become more equal. These scenarios are illustrative or technical in the sense that we do not envisage policies or mechanisms to bring about more equality in the outcomes studied. Rather, the scenarios not only act as a robustness test, but also are informative of the different specific mechanisms driving the GPG and of country-specific differences in how they operate. Finally, they show the speed—or lack thereof—by which the various changes affect the GPG.

We simulate two sub-scenarios. In the first sub-scenario, the *Equal employment* scenario, we impose gender equality in labour market participation, unemployment, and employment rates by age category.[20] Rates among inactive persons in socio-economic states where pension rights are accrued (e.g., disability) are also equalised. In the second sub-scenario, in addition to equalising the employment rates, part-time employment rates and hourly pay are also equalised across genders (the *Equal employment and Wages* scenario). Figure 11 shows the results of these scenarios together with the Base (AWG) scenario.

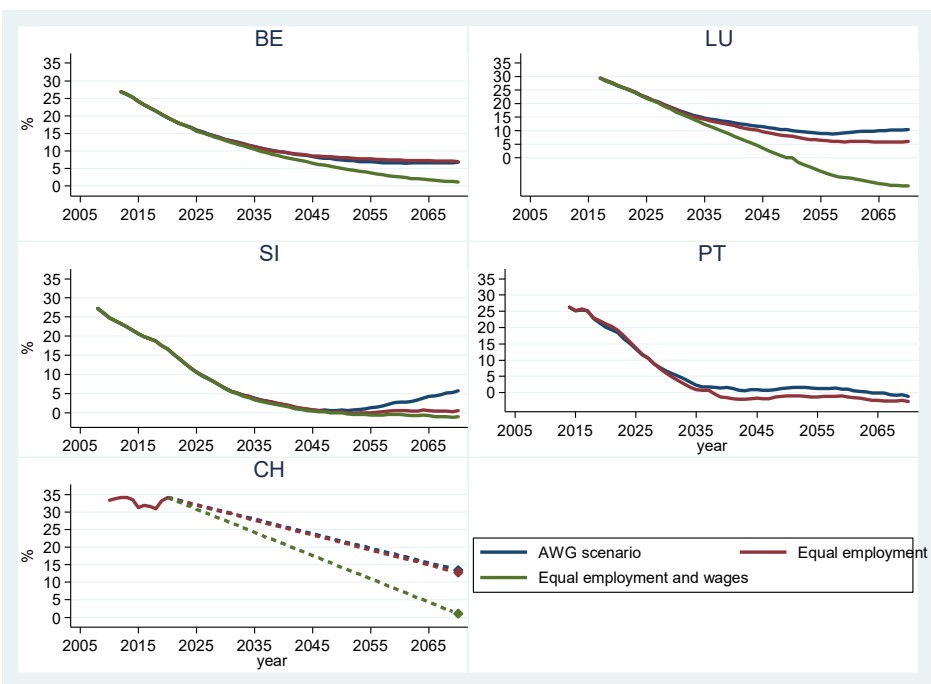

**Figure 11.** Impact on the GPG of gender convergence in employment and pay (2008–2070). Source: Projections by MIGAPE country teams. Note: see Figure 7.

The combined impact of equalising both the employment and earnings between men and women is shown by comparing the Equal employment and Wages scenario with the Base (AWG) scenario. These differences are negligible in Slovenia and Portugal, small in Belgium and considerably larger in Luxembourg and Switzerland. The main effect is caused by the equalisation of part-time employment rates and hourly pay, while the difference between the Base scenario and the Equal Employment scenario suggests that the further reduction of differences between men and women in their employment rate is of lesser importance in the projection of the GPG, and far smaller than the impact of the part-time work gap and wage gap.

Furthermore, we see that the differences between the various scenarios appear only from the second half of the simulation period on. One reason for these gradually increasing effects of the two scenarios is that the equalisations of employment and wages (from 2020) will have only partly affected the population of retirees by 2050. This is particularly

pronounced in the case of Slovenia, where the equal labour market status scenario has no effect at all. The employment rates for the age groups above 35 years are approximately equal already in the AWG scenario, and while the employment rates are projected to be lower for younger women than men in the AWG scenario, equalising them will only show up in the pension outcomes after 2050. Moreover, the gender pay gap in Slovenia is not very wide—around 8%—to start with, and wage equalisation affects the pension assessment base, which is calculated as the average net wage from the best 24 consecutive years, only very slowly. Finally, in Luxembourg, the GPG in the scenario with equalised employment and wage rates will become negative after 2045. The reason for this is twofold and basically follows the reasoning explained in the discussion of the GPG variant including zero pensions. First of all, we argued there that Luxembourg has a minimum contributory period of 10 years, which would affect women still more than men, even at the simulation horizon. However, in this variant with equal work rates and pay, this will no longer be the case, so in Luxembourg the difference for the GPG between the base and equal employment scenarios would be even greater than in the other countries. Furthermore, in the discussion of Figure 9, we argued that the negative impact of the survivors' benefits on the GPG would be reduced. Yet, it would remain more important than in other countries, partly because the reduction of the survivor's benefit in case of an old-age benefit is less severe than in other countries (see Section 3). As a result of the equalising of the work rates and pay, the old-age pensions of men and women would converge further and the GPG would continue decreasing near the simulation horizon. The remaining survivors' benefits would then drive the GPG below zero.

In Belgium, the effect of higher employment rates of women when equalising employment rates is moderated by two effects. Equalisation implies a lower rate of women working as civil servants and a higher proportion being self-employed. Both changes tend to reduce the pension of women because civil servants enjoy a relatively more generous pension system, whereas the opposite is the case for people who are self-employed. Equalization of wages, in addition to equal employment rates, has a larger effect in 2050 and will lead to a closing of the Gender Pension Gap in the long run when fully affecting all retirees. In Belgium and Luxembourg, part-time working rates are also set equal between women and men in the equal-wage scenario, and this also makes an important contribution to the closing of the Gender Pension Gap. Therefore, these results suggest that the gender difference in average pensions may well be nearly eliminated in the next decades in Slovenia and Portugal. In Belgium and particularly in Luxembourg, this will not be the case unless part-time work rates will converge between men and women as well.

In Switzerland, the effect of equal labour market participation has only a marginal impact on the GPG, since the employment rates of women are comparatively high and the differences across gender are already small. Hence, the projected GPG will decline only marginally. In the scenario with equalised employment and wage rates, the GPG is reduced significantly and will essentially disappear. Hence, the simulated GPG in Switzerland follows a similar trend to the one in Belgium and Luxembourg.

## 6. Discussion

Current and future Gender Pension Gaps depend on differences between men and women in the prevalence of part-time work, unemployment rates, withdrawals from the labour market, and the pay gap. These add up in their impact on pension benefit during retirement.

The main goal of the current study is to investigate how the Gender Pension Gap in statutory pensions will develop under the labour market assumptions underpinning pension expenditure projections in the Ageing Report produced by the Working Group on Ageing Populations and Sustainability (AWG) of the Economic Policy Committee (EPC) of the European Union.

We find that the Gender Pension Gap will fall significantly in the four EU countries over the next two decades. In Slovenia and Portugal, the gap will be close to five percent

already in 2030 and will have essentially disappeared in 2040 (although simulations for Slovenia suggest that it starts to increase again near the simulation horizon of 2070). In Belgium and Luxembourg, the gap is reduced to seven and five percent in 2050, respectively, more than two-thirds below the 2020 level, but it is not projected to decline beyond that in the reference scenario. The projected long-term decline in the Gender Pension Gap is partly due to the further reduction in the gender employment gap projected by the AWG. However, even when current labour market outcomes would continue in the future without further equalisation, the Gender Pension Gap would fall nearly as much. In Switzerland, the reduction will be smaller as the gap will be reduced to 13 percent, which is about a third of the 2020 levels. This reduction is driven by an increase in the work intensity of women and a reduction in the wage gap. To eliminate the GPG, a more equal distribution of part-time work and the eradication of the gender pay gap would be required.

Currently, in Belgium, Luxembourg, and Portugal, the Gender Pension Gap would be much larger without survivors' benefits, and in the two latter countries, this impact will persist over time. Only in Slovenia is the impact of survivors' pensions on the Gender Pension Gap already small today.

We also assess how the development of the gender pension gap would change under assumptions of labour market equality between men and women. To this end, we impose gender equality in labour market participation, unemployment and employment rates, part-time employment rates, and hourly pay by age category from 2020 on. The combined impact of both these scenarios in 2050 is negligible in Slovenia (where there is a near-equal situation in part-time work rates today), small in Portugal, larger in Belgium (where various labour market developments counteract) and considerable in Luxembourg and Switzerland. These results suggest that especially the future part-time work rates and the pay gaps will prevent the Gender Pension Gap from being eliminated in Belgium, Luxembourg, and Switzerland, and this is contrary to Slovenia and Portugal. Furthermore, all simulations show that developments in labour market and pay affect the GPG only very gradually.

The issue of the future social adequacy of pension benefits has been gaining increasing attention among policy-makers—as evidenced by the European Commission's "Agenda for Adequate, Safe, and Sustainable Pensions", which states that "Addressing pension adequacy and sustainability therefore requires a mix of pension and employment policies aimed at tackling gender differences in pension incomes" (European Commission 2012, p. 12). Based on statistics on current gender pension gaps, the latest Pension Adequacy Report (European Commission and Social Protection Committee 2021b, p. 105) concludes that the Gender Pension Gap is closing, albeit slowly. Our results suggest that this process will continue, but that, given existing pension systems, a more equal distribution of part-time work rates and the eradication of the gender pay gap would be required to eliminate the Gender Pension Gap in statutory pensions.

**Author Contributions:** Conceptualisation, G.D.; methodology, G.D. and K.V.d.B.; validation, G.D., K.V.d.B., T.K., N.B., N.K., P.L., A.M. and N.S.; formal analysis, G.D., K.V.d.B., M.B., T.K., N.B., N.K., P.L., A.M. and N.S.; data curation, G.D., K.V.d.B., M.B., T.K., N.B., N.K., P.L., A.M. and N.S.; writing-original draft preparation, G.D., K.V.d.B. and M.B.; writing-review and editing, G.D., K.V.d.B., M.B., T.K. and N.S.; project administration, G.D., K.V.d.B. and M.B.; funding acquisition, G.D., K.V.d.B. and P.L. All authors have read and agreed to the published version of the manuscript.

**Funding:** This research was co-funded by the Rights, Equality and Citizenship Programme of the European Union (2014–2020) via Grant Agreement no. 820798. The content of this report represents the view of the author only and is his/her sole responsibility. The European Commission does not accept any responsibility for use that may be made of the information it contains.

**Informed Consent Statement:** Not applicable.

**Acknowledgments:** The authors thank three anonymous reviewers for useful comments, which have improved the article.

**Conflicts of Interest:** The authors declare no conflict of interest.

## Appendix A

**Table A1.** Average replacement rate; legal current and projected retirement ages (following current legislation).

| Country | Aggregate Replacement Rate (1) | | Retirement Ages | | |
|---|---|---|---|---|---|
| | M | F | Early (2) | Normal (3) | Future (3) |
| EU27 | 0.56 | 0.53 | | | |
| BE | 0.47 | 0.46 | 63 | 65 | 67 |
| LU | 0.99 | 0.95 | 62 | 62 | 62 |
| SI | 0.43 | 0.42 | 60 | 62 | 62 |
| PT | 0.67 | 0.64 | 62 | 65.3 | 68 |
| CH | 0.55 | 0.57 | 63 | 65 (M), 64 (F) | 65 (M), 64 (F) |

(1) Eurostat's "Aggregate replacement ratio for pensions (excluding other social benefits) by sex (online data code: TESPN070) last update: 11 May 2022 23:00. Note that the comparatively low replacement rates in Slovenia are partially because pensions in Slovenia are net of taxes. Likewise do special tax arrangements for pensions in Belgium ensure a comparatively low tax burden on these incomes; (2) OECD (2021), Table 3.5." p. 131; (3) OECD (2021), eISSN: 2077–7760, doi:10.1787/pension-data-en OECD database "Pensions at a Glance: Design of pension systems" shows the current and future retirement age for a male or female person who entered the labour force at age 22 in 2020.

## Appendix B. A Discussion of Dynamic Microsimulation of Pensions

Denote the pension benefit in the first year of retirement $T$ as $pb_T$. This can be written as

$$pb_T = G_T \left( \sum_{t=T-x}^{T-1} F_T(y_t) \right)$$

with $y_t$ being the income or revenue at time $t$, $F()$ denoting how each of these income bases is used in the calculation of the pension (for example the annual accrual rate, or floors and ceilings for $y$; given regulations at $T$), while $G()$ denotes how the income base as a whole translates to a pension. This includes the impact of actuarial corrections (bonus or malus for longer or shorter careers), minimum or maximum benefits, etc. Note that for an old-age pension, the income base $y_t$ will be the actual or fictitious income or revenue of the individual itself, but in the case of the survivors' pension, it represents the pension base of the deceased partner. For on-going pensions for periods $t > T$, $pb_t$ can be written as $pb_{t>T} = H_{t>T}(pb_{t-1;t \geq T})$, where, again, the previous value $pb_{t-1}$ can denote the pension benefit of the individual itself, or of the deceased partner. The function $H()$ denotes the indexation regime at T that the pension benefit is subject to at t (which can be different for different pension components), or the "translation" from the old-age pension benefit of the surviving partner into a survivors' benefit. Functions $G()$, $F()$ and $H()$ reflect the pension regulations in force.

Assuming $n - 1$ states that are relevant to the pension system, and one state ($n$) where there is no pension accrual, the input of the pension system in each period is $y = i_1 S_1 + i_2 S_2 + \ldots + i_n 0$. This can be the salary if one works, or a fictitious salary (like in Belgium) if one is unemployed or disabled, or, of course, 0 if one occupies an inactive state $n$. The variable $i_{tx} = [1, 0]$ reflects whether the individual occupies the state $x = (1, 2, 3, \ldots, n)$ at time $t$.

More generally, a discrete-time dynamic microsimulation model works as follows. Denote $I_t$ the vector of all the labour market, social security—and inactivity states at $t$. Hence $I_t = [i_1, i_2, \ldots, i_n]_t$. Seeing over the sample at $t$, $P(I_{tx})$ represents a vector of incidence rates at $t$. The purpose of the model is to simulate transitions of individuals in such a way that the expected value of the resulting incidence rates the next period $t + 1$ equals an exogenous vector of probabilities $PX(I_{(t+1)})$. In the simplest form, we draw a random number u from a uniform distribution, and the individual will occupy state $i_{(t+1)x}$ if $u < PX(I_{(t+1)})$.

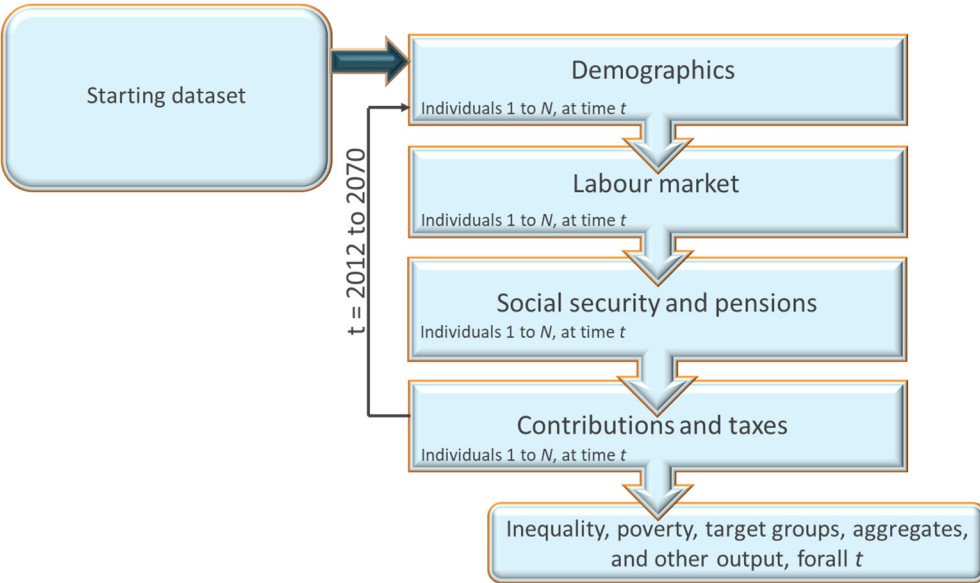

**Figure A1.** General block recursive structure of a dynamic MSM.

This approach of course ignores a wealth of information, including the current vector $I_t$, in the starting data or generated by the simulation process. We first discuss what information is available for simulation at any period. Next, we will discuss how this information can be taken into account in the simulation process.

Assuming a discrete-time model, as we have implicitly done so far, the information that is available at any point in the projection t depends on the structure of the model. Figure A1 above shows the typical block-recursive structure of a dynamic microsimulation model. In many cases, simulation is done using discrete time, mostly a year. This means that the various processes are ordered in a specific sequence in any period, the available information might comprise the vector $I_{q<t} = [I_1, I_2, \ldots , I_n]$ for each period $q$ that precedes $t$, and derived information, like the duration of any state of $I$ over time. However, it also includes contemporary information from blocks of the model that are earlier in the sequence of processes. In Figure A1 above, demographic processes take precedence in each simulation round. This means that this information is available for processes further down the sequence list. In the order as shown in Figure A1, for example, demographic changes at time t can affect the labour market situation at time $t$, but not the other way around (though the latter can affect demographic transitions at time $t + 1$).

So, denote $K(t + 1)$ all (relevant) information available at simulation of $I_{(t+1)}$, with $I_t$ included. This is used in the simulation through logistic regression equations, from which an estimate of their a priori risk of staying in or moving to a state $I_{(t+1)}$ is derived. In case of alignment through sorting, the probability $exp(K)/(1 + exp(K)) + \varepsilon$ is used to rank individuals to their a priory risk of them occurring in state $I_{(t+1)}$, relative to all other n individuals in the risk set. Then, the individuals with the highest rank are selected to occur in that specific state.

**Notes**

[1] The Gender Pension Gap (GPG) measures the relative difference between the pensions of women and men. In a general form, the GPG($l$, $x$) can be written as $1 - \frac{l(x)_f}{l(x)_m}$; usually l is the mean of the variable of interest, x, e.g., gross pension income, though l can be any measure of location.

[2] The official name is the Working Group on Ageing Populations and Sustainability of the Economic Policy Committee.

[3] Further details can be found in the national reports: Dekkers and Van den Bosch (2021), for Belgium; Liégeois (2021), for Luxembourg; Moreira and Wall (2021), for Portugal; Kump and Stropnik (2021), for Slovenia, and Kirn and Bauman (2021), for Switzerland.

4    Gender pay gap in unadjusted form in industry, construction, and services (except public administration, defence, compulsory social security). Source: Eurostat, EARN_GR_GPGR2. This indicator reflects the difference between average hourly earnings of male and female employees working in firms with at least 10 employees, expressed as a percentage of the former.

5    Indeed, women already are more likely than men to provide informal care (OECD 2020, p. 3) and they provide more hours of informal care. As informal care is often accompanied by a reduction or abandonment of professional activity by the caregiver (Ciccarelli and Soest 2018; European Commission and Social Protection Committee 2021a, pp. 83–84), the gender difference in informal care-responsibilities adds to the Gender Pension Gap (Bettio et al. 2013; Burkevica et al. 2015).

6    https://ec.europa.eu/eurostat/web/products-eurostat-news/-/DDN-20200207-1. See also the two most recent Pension Adequacy Reports (European Commission and Social Protection Committee 2018, 2021b), and the 2019 Report on Equality between Women and Men in the EU (European Commission 2019, Figure 6, p. 24).

7    Third-pillar private pensions are not discussed in this article.

8    https://www.bfs.admin.ch/bfs/de/home/statistiken/soziale-sicherheit/berufliche-vorsorge/einrichtungen-versicherte.html (accessed on 8 July 2022).

9    https://www.bfs.admin.ch/bfs/de/home/statistiken/soziale-sicherheit/berichterstattung-altersvorsorge/indikatoren-altersvorsorge/zugang-system-alterssicherung.html (accessed on 8 July 2022).

10    This section is based on European Commission and Social Protection Committee (2021c) for the four EU countries, and on Kirn and Bauman (2021) for Switzerland. Note that the state of the various pension systems are being included as they were up to early 2021, analogous to the projections of the AWG. This includes all forward-looking measures that were legislated at the time of our modelling. An example of the latter are the future increases of the pensionable age in Belgium and Portugal and gender-neutral pension calculation rules in Slovenia.

11    Self-employed people and other non-insured persons can contribute voluntarily.

12    A special arrangement in the Swiss first-pillar pension is that years of care for children below 16 years are counted as contributory years at a rate of three times the minimum pension.

13    See European Commission (2021b, Figure 30, p. 77). Figures not available for Switzerland.

14    A particular issue in this context is the decision to retire. In the Slovenian model, individuals most often retire as soon as they are eligible, but they can decide to work up to 3 years longer (based on the probability). In the other models, however, the retirement decision before the statutory retirement age (i.e. early retirement) is mostly the negative result of the labour market equations, i.e., follows from not remaining in the labour market. Once an individual ceases to work at an older age, which is the result of the combination of behavioural equations and alignment tables by age and gender, then the a priori risk of entering into one of the alternative states (unemployment, disability, etc.) becomes very low if he or she is eligible for retirement, with the result that he or she will very likely retire. At the statutory retirement age, all individuals will by definition enter retirement.

15    Figures for 2019. Eurostat, table "lfsa_eppgacob".

16    Each month counts towards the minimum contributory period as soon as 64 h of work have been registered. Furthermore, "surplus" hours worked can be transferred from one month to the next. Therefore, if working 20% of a full time job, one month of every second month counts as a contributory period. Finally, contributory periods include unemployment, registered care periods, and maternity leave.

17    In 2018, expenditures on old-age and survivor pensions were equal to 9.6% and 1.6% GDP in the EU-27 as a whole (European Commission and Social Protection Committee 2021b, p. 35).

18    The impact of survivors' pension could not be determined in the model for Switzerland.

19    Do note that this scenario does *not* mean that individuals remain in the state that they occupy in 2021, but rather that the proportional sizes of the various labour market states remain at their 2021 levels. In the reference scenario, the proportional sizes of the various (labour market) states by age and gender change over time. For example, the activity rate among women increases towards that of men; unemployment rates decrease, etc. All these developments are blocked in the constant scenario. However, even though the proportional sizes of the various categories remain constant, individuals still move from one state to another.

20    For Portugal, only private employment and self-employment rates are equalised. The relative shares of public workers and civil servants remain unchanged (Moreira and Wall 2021).

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
