# Peer review of "How Do Gendered Labour Market Trends and the Pay Gap Translate into the Projected Gender Pension Gap? A Comparative Analysis of Five Countries with Low, Middle and High GPGs"

_socsci, doi:10.3390/socsci11070304_

Round 1
Reviewer 1 Report
See .pdf file

Reviewer 2 Report
The first thing is to congratulate the authors for the work, it is current and provides a lot of knowledge.
In my opinion, the size of the keywords must be reduced, as well as including at least one in which the word “genre” appears, it cannot be that this work does not appear in a search by keywords in which “gender” is searched ”, the keyword “gender gap” would be fine. The abstract must also clearly express what the main conclusion of the work is.
The graphs should all have the same format, figures 1, 2, 3…, they all have a different format. At least provide a same size.
They must clearly define the main objective of the work in the introduction.
Throughout the work the regulation of pensions in the countries studied is explained, it does not seem that this information has much relevance for the objective, since in all the countries analyzed there is no gender distinction in that regulation.
No research hypothesis is clearly stated, which means that the results remain up in the air, and therefore they cannot be taken for granted, they must state and list the research hypotheses, detailing them since they do not appear in the work.

Reviewer 3 Report
It would be interesting to explain, why only several selected countries analyzed? What is the rationale to include only five of them? Where there other examples of countries with low and high GPG that could have been included? I suppose it is a data driven research, so that might be the reason for the selection.
Can you provide break down of GPG percentage for all EU countries in the introduction, in order to help reader understand which countries have low, medium or high GPG?
